# Increased Prevalence of Lassa Fever Virus-Positive Rodents and Diversity of Infected Species Found during Human Lassa Fever Epidemics in Nigeria

Anise N. Happi,[a] Testimony J. Olumade,[a,b] Olusola A. Ogunsanya,[a] Ayotunde E. Sijuwola,[a] Seto C. Ogunleye,[c] Judith U. Oguzie,[a,b] Cecilia Nwofoke,[d] Chinedu A. Ugwu,[a,b] Samuel J. Okoro,[d] Patricia I. Otuh,[e] Louis N. Ngele,[d] Oluwafemi O. Ojo,[f] Ademola Adelabu,[g] Roseline F. Adeleye,[f] Nicholas E. Oyejide,[a] Clinton S. Njaka,[h] Jonathan L. Heeney,[i] Christian T. Happi[a,b]

[a]African Centre of Excellence for Genomics of Infectious Disease, Redeemer's University, Ede, Osun, Nigeria
[b]Redeemer's University, Ede, Osun, Nigeria
[c]University of Ibadan, Ibadan, Oyo, Nigeria
[d]Alex Ekwueme Federal University Teaching Hospital, Abakaliki, Ebonyi, Nigeria
[e]Michael Okpara University of Agriculture, Umudike, Abia, Nigeria
[f]Federal Medical Centre, Owo, Ondo, Nigeria
[g]Rufus Giwa Polytechnic, Owo, Ondo, Nigeria
[h]Zeal Veterinary Services, Enugu, Enugu, Nigeria
[i]Lab of Viral Zoonotics, University of Cambridge, Cambridge, United Kingdom

**ABSTRACT** The dynamics of Lassa virus (LASV) infections in rodent reservoirs and their endemic human caseloads remain poorly understood. During the endemic period, human infections are believed to be associated with the seasonal migration of *Mastomys natalensis*, thought to be the primary reservoir that triggers multiple spillovers of LASV to humans. It has become imperative to improve LASV diagnosis in rodents while updating their prevalence in two regions of Lassa fever endemicity in Nigeria. Rodents (total, 942) were trapped in Ondo (531) and Ebonyi (411) states between October 2018 and April 2020 for detection of LASV using various tissues. Overall, the LASV prevalence was 53.6%. The outbreak area sampled in Ondo had three and two times higher capture success and LASV prevalence, respectively, than Ebonyi State. This correlated with the higher number of annual cases of Lassa fever (LF) in Ondo State versus Ebonyi State. All rodent genera (*Mastomys*, *Rattus*, *Crocidura*, *Mus*, and *Tatera*) captured in both states showed slightly variable LASV positivity, with *Rattus* spp. being the most predominantly infected (77.3%) rodents in Ondo State versus *Mastomys* spp. (41.6%) in Ebonyi State. The tissues with the highest LASV positivity were the kidneys, spleen, and testes. The finding of a relatively high LASV prevalence in all of the rodent genera captured highlights the complex interspecies transmission dynamics of LASV infections in the reservoirs and their potential association with increased environmental contact, as well as the risk of zoonotic spillover in these communities, which have the highest prevalence of Lassa fever in Nigeria.

**IMPORTANCE** Our findings show the highest LASV positivity in small rodents ever recorded and the first direct detection of LASV in *Tatera* spp. Our findings also indicate the abundance of LASV-infected small rodents in houses, with probable interspecies transmission through vertical and horizontal coitus routes. Consequently, we suggest that the abundance of different reservoir species for LASV may fuel the epizootic outbreaks of LF in affected human communities. The high prevalence of LASV with the diversity of affected rodents has direct implications for our understanding of the transmission risk, mitigation, and ultimately, the prevention of LF in humans. Optimal tissues for LASV detection in rodents are also presented.

**KEYWORDS** Nigeria, Lassa fever, Lassa virus, small rodents, infected tissues, Lassa diagnosis

Address correspondence to Anise N. Happi, happia@run.edu.ng, or Christian T. Happi, happic@run.edu.ng.

The authors declare no conflict of interest.

Lassa fever (LF), caused by the arenavirus Lassa virus (LASV), is a zoonotic hemorrhagic disease which, since its first discovery in 1969, has been found to be endemic to the Western region of Africa, particularly Nigeria (1, 2), Guinea (3), Sierra Leone (3), Liberia (4), and Mali (5). The Nigeria Centre for Disease Control (NCDC) cumulative national LF situation update from January to 15 August 2021 reported 354 confirmed cases with 73 fatalities (case fatality rate [CFR], 20.6%) (6). During the 2018 outbreak of LASV in Nigeria, genome sequencing and phylogenetic analysis revealed that the outbreak was sustained by an increased transmission from rodent reservoirs with relatively few nosocomial transmissions (2). As first reported by Monath and colleagues (7), *Lassa mammarenavirus* is primarily transmitted to humans from the African multimammate mouse (*Mastomys natalensis*), the natural host reservoir. Since then, LASV has also been found in other rodent species, such as *Mastomys erythroleucus*, *Hylomyscus pamfi*, *Rattus rattus*, and *Mus musculus* (1, 8, 9). It remains unknown if each of these species separately maintains LASV over generations through horizontal or vertical transmission cycles, or as a consequence of interspecies transmission due to territorial overlap and mixing, or if some of these species naturally maintain cycles of LASV infection independently over generations.

Numerous studies have reported the prevalence of LASV by reverse transcriptase PCR (RT-PCR) between 11.3% and 50% in a few rodent species from a number of countries in West Africa using blood (1, 3, 8, 10), single tissue (5, 10, 11), or pooled tissue (12) samples. A seroprevalence of 11% was also reported with the detection of LASV-specific antigen (using blood and spleen homogenates) by Demby et al. (13). In Sierra Leone, however, Monath et al. (7) reported a LASV prevalence of 50% in *M. natalensis* using tissue culture from a pool of tissues (heart, muscle, lung, kidney, spleen, salivary gland, intestine, and Peyer's patches). Leski et al. (11) reported a LASV prevalence of 19.16% by RT-PCR using spleen tissue only, while Safronetz et al. (5) reported a 19.4% LASV prevalence by serology and RT-PCR from blood and liver samples from *M. natalensis* in Mali. In Benue, Plateau, and North-Eastern states in Nigeria, Wulff et al. (14) reported a 3.3% prevalence of LASV in *M. natalensis*, *R. rattus*, and *Mus minutoides* by virus isolation and complement fixation using blood, urine/bladder, and pooled organs (liver, lung, kidney, and spleen). In southern Nigeria, a LASV prevalence of 4.8% was reported in *Mastomys erythroleucus*, 41.6% in *Hylomyscus pamfi* (1), and 1.6% in *R. rattus*, *M. musculus*, and *M. natalensis* (8) by RT-PCR of blood samples. However, in the wake of increasing outbreaks of human LF cases in Nigeria, there seemed to be a disparity between the number of rodent infections and the human LF case rates in affected communities. These differences in findings pointed to a need for standardized rodent sampling techniques, for further detailed understanding of the true prevalence and infection dynamics of LASV in animal reservoirs. Thus, this study was designed to evaluate the relationship between human LF cases and LASV carriage by small rodent populations in two major LF hot spots in Nigeria. To accomplish this, we undertook a comprehensive LASV prevalence survey in different small rodents in two major regions of endemicity in Nigeria. Through this survey and analysis of different types of samples, we identified the optimal tissues for standardizing LASV detection in small rodents. Furthermore, we identified a high incidence of LASV and a broad spectrum of different rodent species infected with LASV in different outbreak settings.

## RESULTS

**Small rodent capture success. (i) Overview of capture rates in Ondo and Ebonyi States.** From 2018 to 2020, over 942 small rodents were trapped (from 262 homes and 4 local fields) and sampled from two southern states where LF is endemic in Nigeria. In Ebonyi State, 531 rodents were trapped between October 2018 and February 2020, and in Ondo State, 411 were collected between August and October 2019. The capture rate in Ondo State was higher (18.6%) than that in Ebonyi State (5.8%) (Fig. 1), although this difference was not statistically significant ($P > 0.05$). A total of four small rodent genera were identified (*Mastomys* [54%], *Rattus* [39.7%], *Crocidura* [5.8%], and

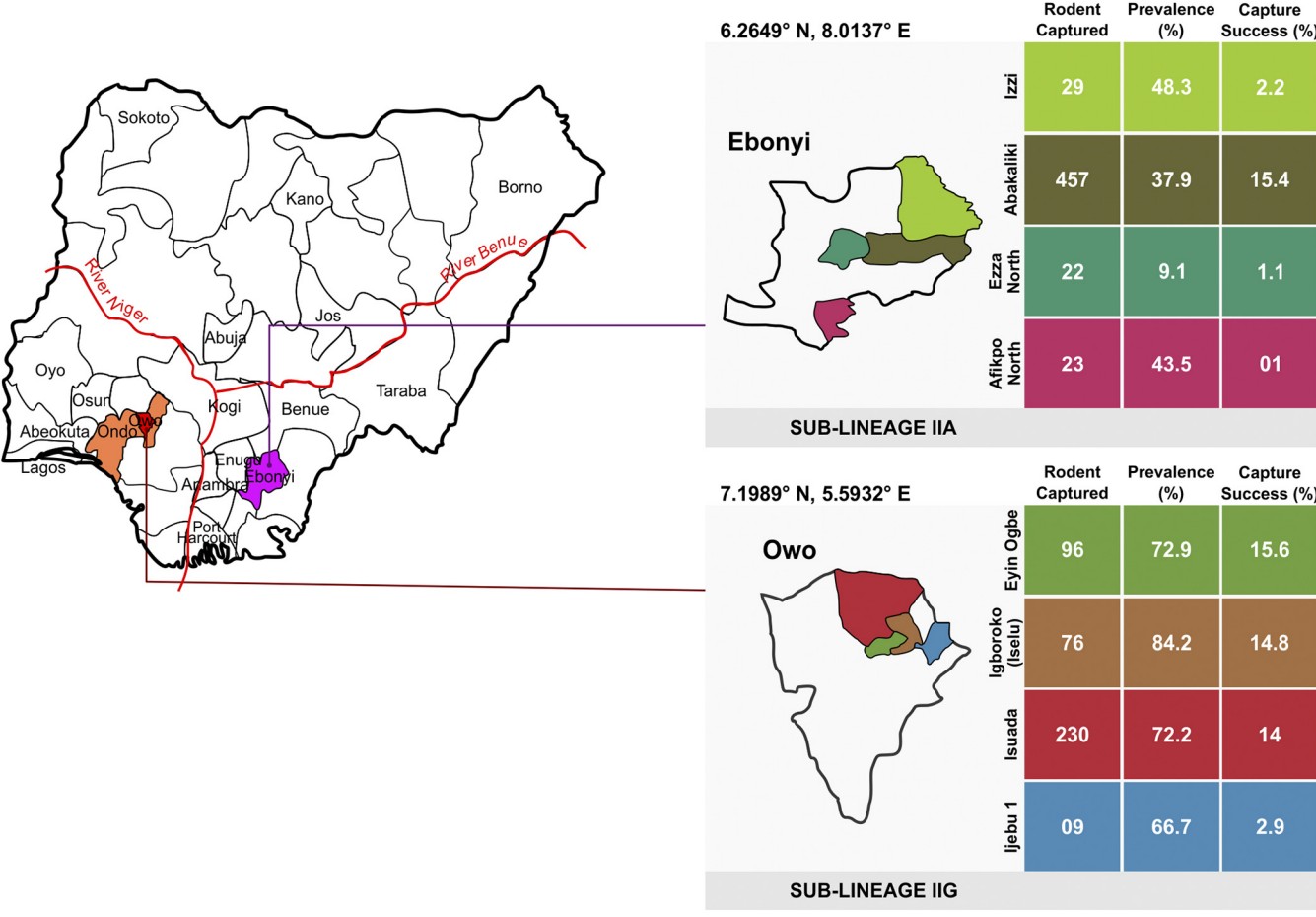

**FIG 1** Map of Nigeria showing Ondo (Yoruba tribe) and Ebonyi (Igbo tribe) states, the trapping locations in each state, and the corresponding capture success and prevalence from the trapping locations.

*Mus* [0.5%]) in Ondo State, while five small rodent genera (*Mastomys* [51.6%], *Crocidura* [26.4%], *Rattus* [18.8%], *Mus* [0.6%], and *Tatera* [0.2%]) were identified in Ebonyi State. *Mastomys* spp. had the highest capture rates in both Ondo (9.3%) and Ebonyi (3%) states (Table 1). The second most prevalent species was *Rattus* spp. in Ondo State, while in Ebonyi State, it was *Crocidura* spp. (Table 1). Thirteen other small rodents captured in Ebonyi State were unidentified.

(ii) **Capture rates in local government areas.** In Ondo State, small rodents were trapped from only one local government area (LGA) (Owo) comprising several communities. The capture success was the same as presented for Ondo State above. However, in Ebonyi State, rodents were captured from four LGA (Izzi, Abakaliki, Ezza North, and Afikpo North), with Abakaliki having the highest capture success (15.4%) (Fig. 1).

Of the four communities in Owo Local Government Area, Ondo State, Ehin-Ogbe had the highest rodent capture rate (15.6%), followed by Iselu (14.8%) (Table 1). *Mastomys* spp. were the most frequently trapped species in most communities, except at Ijebu-Owo, where *Crocidura* spp. (1.9%) were most prevalent (Table 1). In Ebonyi State, traps were set in one community in each local government region. The capture success was highest in the community of Agbaja in Abakaliki LGA (15.4%) (Table 1).

(iii) **Capture rates in rural and urban areas.** All communities where rodents were trapped in Ondo State were urban, hence the high capture rate of 18.6%. However, in Ebonyi State, the capture rate in the rural area (Izzi) was 6.8%, while in the urban areas (Abakaliki, Ezza North, and Afikpo North local government areas), the capture rate was 6%.

(iv) **Capture rates in homes and fields.** In Ondo State, a rodent capture rate of 17% was recorded in homes, while the rate in fields was 0.1%. In Ebonyi State, rodent

**TABLE 1** Capture success of small rodent species by community

| Rodent species | Capture rate (%) for: | | | | | | | | | |
| | Owo LGA communities | | | | | Ebonyi State LGAs | | | | |
| | Ijebu-Owo | Isuada | Iselu | Ehin-Ogbe | Total | Izzi | Abakaliki[a] | Ezza North | Afikpo North | Total |
| *Mastomys* spp. | 0.3 | 6.9 | 8 | 10.9 | 9.3 | 1 | 7.8 | 0.6 | 0.7 | 3 |
| *Crocidura* spp. | 1.9 | 0.7 | 0.4 | 0.8 | 1 | 0.3 | 4.3 | 0.2 | 0.2 | 1.5 |
| *Rattus* spp. | 0.6 | 6.3 | 6.4 | 3.9 | 6.8 | 0 | 3.2 | 0.1 | 0.04 | 1.1 |
| *Mus* spp. | 0 | 0.1 | 0 | 0 | 0.1 | 0.2 | 0 | 0 | 0 | 0.03 |
| *Tatera* spp. | 0 | 0 | 0 | 0 | 0 | 0.1 | 0 | 0 | 0 | 0.01 |
| Unidentified | 0 | 0 | 0 | 0 | 0 | 0.1 | 0.1 | 0.1 | 0.04 | 0.1 |
| Total | 2.9 | 14 | 14.8 | 15.6 | | 2.2 | 15.4 | 1.1 | 1 | |

[a]Abakaliki is a combination of two local government areas (Ebonyi and Abakaliki).

capture rates of 5.8% and 0.03% were recorded in homes and fields, respectively. In Ondo State, all rodent species identified were trapped in homes (Table 2). The most frequently trapped species in homes were *Mastomys* spp. (9.2%), *Rattus* spp. (6.8%), and *Crocidura* spp. (1%), and the least frequent was *Mus* spp. (0.08%) (Table 2). However, *Mastomys* spp. and *Crocidura* spp. were the only species trapped in fields (Table 2). In Ebonyi State, in addition to *Crocidura* spp., *Mus* spp. and *Tatera* spp. were only trapped in fields/on farms (Table 2). In homes in Ebonyi State, *Mastomys* spp. (3%), *Crocidura* spp. (1.5%), and *Rattus* spp. (1.1%) were the only identified species trapped (Table 2).

**Prevalence of LASV infection among small rodents. (i) Overview of prevalence in Ebonyi and Ondo States.** The prevalence of LASV in small rodents within the sampling period was calculated as such: the total number of LASV-positive rodents divided by the total number of rodents tested within the designated community/site. Of all 942 rodents captured from both states, 505 (53.6%) were positive for LASV in at least one of the tissues tested. In Ondo State, the LASV prevalence among small rodents was twice as high (74.5%) as in Ebonyi State (37.5%). Interestingly, although the LASV prevalence rate differed significantly, PCR evidence of infection could be found in the different varieties of rodents captured. In Ondo State, the LASV prevalence varied among the rodents captured, with *Rattus* spp. being the most frequently infected (77.3%). In contrast, in Ebonyi State, the LASV prevalence was lower and moderately variable, with *Mastomys* spp. showing the highest LASV positivity (41.6%) (Fig. 2). Of the 13 unidentified small rodents captured in Ebonyi, 4 (30.8%) were LASV positive.

There was no significant difference in LASV positivity between male and female rodents ($P = 0.252$).

**(ii) Prevalence in local government areas and their communities.** In Ondo State, the LASV prevalence was 74.5% in Owo, the only local government area (LGA) sampled. Iselu was the community in the Owo LGA with the highest LASV prevalence (84.2%) in rodents (Table 3). Other communities showed various species with high LASV infection rates (Table 3). For instance, *Rattus* spp. were the rodents with the highest LASV infection

**TABLE 2** Capture success of rodent species by location

| Rodent species | Capture rate (%) in: | | | |
| | Ondo State | | Ebonyi State | |
| | Homes | Fields | Homes | Fields |
| *Mastomys* spp. | 9.2 | 0.04 | 3 | 0 |
| *Crocidura* spp. | 1 | 0.08 | 1.5 | 0.02 |
| *Rattus* spp. | 6.8 | 0 | 1.1 | 0 |
| *Mus* spp. | 0.08 | 0 | 0 | 0.03 |
| *Tatera* spp. | 0 | 0 | 0 | 0.01 |
| Unidentified | 0 | 0 | 0.1 | 0 |
| Total | 17 | 0.1 | 5.8 | 0.03 |

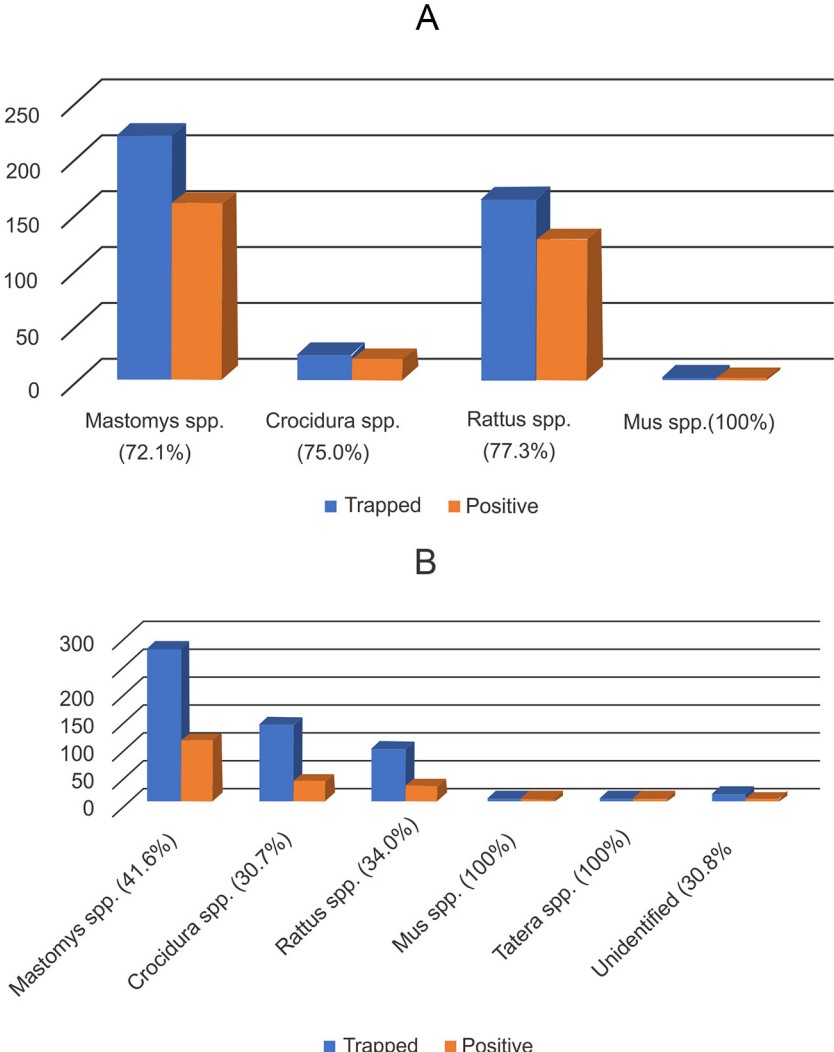

**FIG 2** Frequency distribution of LASV positivity/prevalence in small rodent species trapped in Ondo (A) and Ebonyi (B) states.

rates in the communities of Ijebu-Owo (100%) and Ehin-Ogbe (79.2%), while in Isuada, *Crocidura* spp. (90.9%) had the highest LASV infection rates (Table 3). There was a significant difference in capture success rates and LASV prevalence in small rodents across the communities in Ondo State ($P < 0.05$). In Ebonyi State, however, Izzi Local Government Area had the highest LASV prevalence (48.3%) in small rodents, with *Crocidura* spp. being the most LASV-infected (75%) rodent (Table 3). Although *Tatera* spp. and *Mus* spp. had a prevalence of 100%, very few were also captured (Table 3). Within Ebonyi State, a significant difference in the number of rodents trapped and LASV infected across the communities was also recorded ($P < 0.05$).

(iii) **Prevalence in rural and urban areas.** In Ondo State, the LASV prevalence was 74.5% in urban areas. In Ebonyi State, 48.3% of rodents captured from rural areas were LASV positive, as against 51.7% in urban areas. Comparison of the LASV prevalence in urban areas between Ondo and Ebonyi states showed no statistically significant difference ($P = 0.13$).

(iv) **Prevalence in homes and fields.** In Ondo State, the LASV prevalence was twice as high in homes as in fields (74.8% versus 33.3%), while in Ebonyi State, the LASV prevalence was slightly higher in fields than in homes (40% versus 37.4%). However, there was no significant difference in LASV prevalence between homes and fields for both states ($P = 1.00$ in Ondo and $P = 0.46$ in Ebonyi).

**TABLE 3** LASV prevalence in small rodent species by community in Ondo State (Owo LGA only) and Ebonyi State (four LGAs)

| | LASV prevalence (%) in: | | | | | | | | | |
| --- | --- | --- | --- | --- | --- | --- | --- | --- | --- | --- |
| | Ondo State | | | | | Ebonyi State | | | | |
| | Owo | | | | | Izzi | Abakaliki[a] | Ezza North | Afikpo North | |
| Rodent species | Ijebu-Owo | Isuada | Iselu | Ehin-Ogbe | Total | Onu-Enyim | Agbaja | Oriuzor | Afikpo | Total |
| *Mastomys* spp. | 0 | 67.3 | 87.8 | 71.6 | 72.1 | 30.8 | 43.7 | 15.4 | 41.2 | 41.6 |
| *Crocidura* spp. | 66.6 | 90.9 | 50 | 60 | 75 | 75 | 30.5 | 0 | 25 | 30.7 |
| *Rattus* spp. | 100 | 75 | 81.8 | 79.2 | 77.3 | 0 | 34.4 | 0 | 100 | 34 |
| *Mus* spp. | 0 | 100 | 0 | 0 | 100 | 100 | 0 | 0 | 0 | 100 |
| *Tatera* spp. | 0 | 0 | 0 | 0 | 0 | 100 | 0 | 0 | 0 | 100 |
| Unidentified | 0 | 0 | 0 | 0 | 0 | 37.5 | 0 | 0 | 100 | 30.8 |
| Total | 66.7 | 72.2 | 84.2 | 72.9 | | 48.3 | 37.9 | 9.1 | 43.5 | |

[a]Abakaliki is a combination of two local government areas (Ebonyi and Abakaliki).

The correlation between capture success rates and LASV prevalence in Ondo State was moderately positive ($r = 0.4$), with no statistically significant difference ($P = 0.44$). However, in Ebonyi State, a moderately negative correlation ($r = -0.5$) exists between the capture success rate and LASV prevalence, with no statistically significant difference ($P = 0.28$).

**(v) LASV positivity rate in small rodent tissues.** Analysis of different tissues by PCR showed various degrees of LASV positivity for all tissues tested for both sites. In Ondo and Ebonyi states, the organs with the highest LASV positivity were the spleen, kidneys, testes, and embryos (Table 4), with the kidneys and spleen being the most significantly ($P < 0.01$) LASV-infected organs compared to other tissues, irrespective of the sex of the animal. Hence, they were subsequently chosen for LASV diagnosis. Therefore, we classified a rodent as LASV positive when at least one of the tissues tested positive. Plasma was the least (14.9% in Owo and 5.7% in Ebonyi State) positive tissue (Table 4).

Comparing the frequency of LASV positivity in each tissue type across three species (*Mastomys* spp., *Crocidura* spp., and *Rattus* spp.), we observed that only the spleen had significantly high LASV tropism (Table 5). The spleen of *Crocidura* spp. showed the highest LASV tropism, significantly higher than that in *Mastomys* and *Rattus* species (Table 5).

**TABLE 4** LASV positivity rates by rodent tissue type in Ebonyi and Ondo States

| Location | Organ | No. of rodents tested | No. of rodents positive | Proportion | Percentage |
| --- | --- | --- | --- | --- | --- |
| Ondo | Kidney | 408 | 220 | 0.539 | 53.9 |
| | Spleen | 406 | 226 | 0.556 | 55.7 |
| | Liver | 95 | 37 | 0.389 | 39 |
| | Intestine | 96 | 34 | 0.354 | 35 |
| | Lungs | 96 | 35 | 0.365 | 36.5 |
| | Embryo | 24 | 10 | 0.417 | 41.7 |
| | Testes | 83 | 43 | 0.518 | 51.8 |
| | Brain | 34 | 6 | 0.176 | 17.7 |
| | Bone marrow | 27 | 5 | 0.185 | 18.5 |
| | Plasma | 121 | 18 | 0.149 | 14.9 |
| Ebonyi | Kidney | 508 | 123 | 0.242 | 24.2 |
| | Spleen | 465 | 98 | 0.211 | 21.1 |
| | Liver | 92 | 14 | 0.15 | 15.2 |
| | Intestine | 92 | 11 | 0.119 | 12 |
| | Lungs | 103 | 13 | 0.126 | 12.6 |
| | Embryo | 45 | 13 | 0.289 | 28.9 |
| | Testes | 101 | 34 | 0.336 | 33.7 |
| | Brain | 71 | 14 | 0.197 | 19.7 |
| | Bone marrow | 32 | 6 | 0.187 | 18.8 |
| | Plasma | 53 | 3 | 0.057 | 5.7 |

**TABLE 5** Frequency of LASV positivity in tissues by small rodent species in Ondo and Ebonyi States

| Organ | % positivity in: | | | P value |
| --- | --- | --- | --- | --- |
| | *Mastomys* spp. | *Crocidura* spp. | *Rattus* spp. | |
| Kidney | 34.3 | 29.4 | 35.6 | 0.0815 |
| Spleen | 35.2 | 6.3 | 35.6 | 0.0433 |
| Liver | 37.4 | 17.6 | 31.1 | 0.8829 |
| Intestine | 31.5 | 11.8 | 31.1 | 0.8546 |
| Lung | 31.5 | 11.8 | 24.4 | 0.8423 |

**(vi) Prevalence by year.** The overall LASV infection rate in small rodents was significantly higher ($P < 0.0001$) in Ondo than in Ebonyi State in 2019 (Table 6). In Ebonyi, the LASV prevalence rates in small rodents were 46.4%, 33.9%, and 43.5% in 2018, 2019, and 2020, respectively.

## DISCUSSION

This study reveals the high infection rate of LASV in small rodents from Ondo and Ebonyi states, where LF is endemic in Nigeria. Sublineages IIa and IIb have been shown to circulate in Ondo and Ebonyi states, respectively (2). Although there was a spectrum of LASV-positive rodents captured, there was a positive correlation between the capture success and LASV prevalence among small rodents in Ondo State. Importantly, this study identified the greatest diversity and frequency of infected rodent species in areas of the highest human LF caseloads. This finding suggests not only that higher rodent LASV carriage is a greater risk for human cases but that the dynamics of interspecies transmission among different species of rodents (such as *Rattus* spp.) carry additional risk factors, especially given that these species are highly invasive and cohabit human households, carrying the virus with them. Furthermore, the complexity of intraspecies transmissions in rodents may greatly increase not only the risk of human spillover and disease burden but also the duration of LASV epidemics in affected urban areas. This may be an important underlying factor influencing the fact that Owo, Ondo State, has consistently had the highest human LF case rate in Nigeria in recent years. Ebonyi State showed a negative correlation between the capture success and LASV prevalence in rodents, suggesting that despite the significant rodent population, there were fewer LASV-infected animals and therefore a lower risk of human transmission and infection. This finding translates into the relatively lower rate of LF cases in Ebonyi than in Ondo State over the years (6, 15).

The PCR-positive rodent frequency rate was three times higher in Ondo State than in Ebonyi State, and this correlates with the human LF case rate observed in Ondo State, which was three times higher than in Ebonyi State between 2018 and 2020 (15). Our findings may show a relationship between the prevalence of a highly LASV-positive rodent population and a high LF case rate in humans.

The overall prevalence of LASV in small rodents reported in this study is higher than that previously reported (1, 3, 5, 8, 12, 16). Other studies conducted in Nigeria have revealed the overall LASV prevalence in rodents to be between 1.6% (8) and 4.76% (1). In

**TABLE 6** Prevalence of LASV in small rodent species by state in August to November 2019

| Species | LASV prevalence (%) in: | | P value |
| --- | --- | --- | --- |
| | Ebonyi | Ondo | |
| *Mastomys* spp. | 41.3 | 72.1 | 0.0001 |
| *Crocidura* spp. | 28.2 | 75 | 0.0001 |
| *Rattus* spp. | 26 | 77.3 | 0.0001 |
| *Mus* spp. | 0 | 100 | |
| *Tatera* spp. | 0 | 0 | |
| Unidentified | 20 | 0 | |

other West African countries, studies have also shown overall prevalences of LASV in small rodents between 1.2% and 23.3% (Guinea) (5, 12) and 19.4% and 24% (Mali) (5, 16). The high prevalence of LASV in rodents captured in this study may be due to the comprehensive survey of rodent tissues (in addition to whole blood), contrary to surveys that used only whole blood (1, 8, 9). The use of whole-blood samples from animals for detection of LASV may result in missing the presence of the virus in some animals, as acute-phase viremia is transient in blood, while LASV persists in specific tissues (17). A similar phenomenon has been reported for Ebola (18, 19). Previous findings by Fichet-Calvet et al. (20) indicate that *Mastomys* spp. clear viremia within a period (between 1 and 3 months) significantly shorter than their life span. Thus, the true picture of LASV prevalence in small mammals can only be accurately ascertained by testing both tissues and whole blood. Optimal tissue samples are suggested for standardizing LASV confirmation in rodents but may differ slightly across regions, species, and possibly sublineages. Due to the gap in knowledge about the stage of infection at the time of sampling and the optimal rodent tissues to target, diagnosis may be challenging. Hence, choosing one tissue for diagnosis of LASV in rodents may return a false-negative result. Therefore, for a more accurate LASV diagnosis in small rodents by PCR, we recommend that at least three tissues (kidney, spleen, and testis [in males], liver [in females], or embryo [in gravid does]) be tested, as a single tissue will not be sufficient for a reasonable conclusion.

More than half of the small rodents captured in Ebonyi and Ondo states were *Mastomys* spp., as reported by Bonwitt et al. (21) in Sierra Leone. The various genera of small rodents morphologically identified in the two localities sampled in Nigeria were *Mastomys*, *Crocidura*, *Rattus*, *Mus*, and *Tatera*. Other studies conducted in Nigeria (1, 8, 9) described similar findings. Although *Mastomys* spp. was the reservoir host with the best capture success in both chosen regions, it was not the species with the highest rate of LASV infection in Owo, Ondo State. In Owo (more urban setting), the black *Rattus* spp. rodents had the most LASV infections, followed by *Crocidura* spp. This new finding expands our understanding of the complexity of animal reservoirs and the interspecies dynamics of LASV in different settings. The molecular confirmation of the small rodent species described in this study is reported in detail elsewhere (A. N. Happi et al., submitted for publication). Early reports had described *Mastomys* spp. as the primary species harboring LASV (3, 10, 13, 22). Other species, such as *Mastomys erythroleucus*, *Hylomyscus pamfi*, and more recently, *Rattus* spp. (8), have also been found to be reservoirs of LASV (1, 10, 15).

In this study, *Rattus* sp. rodents were the most predominantly infected (77.3%) in Owo, Ondo State. This has a direct implication for the possible transmission of LASV between *Rattus* spp. and humans, as they are the most abundant in homes in Nigeria (8). *Rattus* is a better climber than most small rodents and is found mostly in roofs/ceilings, actively infesting human houses and thereby creating greater opportunities for infection of humans through various routes.

Although there has been a previous report of LASV-specific antibody (IgG) in *Crocidura* spp. captured in Nigeria (9), to the best of our knowledge, this is the first report of PCR detection of LASV in *Tatera* and *Crocidura* spp. In addition, all small rodent genera herewith identified showed positivity to LASV, suggesting that the virus circulates among small rodents sharing the same environment or ecosystem, with potential maintenance of the virus through frequent interspecies transmission.

The higher trapping success in homes than in fields in both Ondo and Ebonyi states suggests a higher exposure risk to LASV from small rodents in household settings. The abundance of small rodents in homes increases zoonotic transmission risks to humans through aerosol inhalation of virus-laden dust particles, ingestion of food contaminated with rodents' excreta (urine and feces), and handling or in some cases consumption of infected rodents (3, 21, 23). In addition, some villagers in our study sites reported rats nibbling their extremities (e.g., fingers) during sleep, indicating another potential human LASV infection route.

Furthermore, in rural areas in Ebonyi State, small rodents had easy access to homes due to the porous housing type. The poor hygiene and food storage systems attract

rodents into homes, which might be responsible for the higher trapping success and LASV prevalence recorded in rural areas.

LASV in small rodents was widely distributed in various tissues, such as the kidneys, spleen, intestine, liver, lungs, testes, embryos, brain, and bone marrow, confirming the ability of the virus to cause multisystemic disease in humans (17, 24, 25). The detection of LASV in embryonic and fetal samples at nonspecific stages of gestation collected during this study is strongly suggestive of vertical transmission. Our study provides the first documented inference of LASV vertical transmission in a reservoir, initially proposed by Fichet-Calvet et al. (26). Furthermore, LASV detected in the testes of male rats suggests the presence of the virus in semen, with the possibility of transmission by coitus. The detection of LASV in the brain also confirms the ability of the virus to cross the blood-brain barrier, such as with lymphocytic choriomeningitis virus (LCMV) in mice. The presence of the virus in cerebrospinal fluid as observed by Gunther et al. (27) equally supports the assertion that LASV can cross the blood-brain barrier. This observation may explain the pathogenesis of the neurological signs and hearing loss in some severe LF cases (28, 29).

Overall, we found that LASV is best detected in small rodent kidneys, spleen, testes, and embryos and least likely to be found in blood. This is probably due to the relatively short viraemic phase of LASV in the blood. However, subsequent sequestration of LASV in key organs such as kidneys, where there is persistent infection and shedding in the environment, may facilitate transmission. This remains a significant factor in sustaining zoonotic and inter- and intraspecies transmission.

In this study, our observation that trapped rodents tested positive year round might not be unconnected to the year-round reports of LF cases in Nigeria by the NCDC (6) in recent years.

In summary, this study found the highest prevalence of LASV among highly diverse rodents in LF-affected communities reported to date. These findings are correlated with the high Lassa case rates in the studied areas. The identification of a new reservoir species (*Tatera* spp.), as well as the high prevalence of LASV in a wide range of rodents, has direct implications for our understanding of transmission risk, mitigation, and ultimately prevention of LF in humans. The multiple species of LASV-positive rodents in communities of endemicity also suggest the complexity of the reservoir, with great interspecies transmission potential. PCR confirmation of positive testis and embryo samples is indicative of vertical and horizontal transmission. Lastly, testing of multiple tissues is recommended for a comprehensive and standardized rodent surveillance protocol for determining the zoonotic risks of LF in affected communities.

## MATERIALS AND METHODS

**Ethical approval.** Ethical approval for this study was obtained from the institutional review boards (IRBs) at Redeemer's University (Ede, Osun State, Nigeria).

**Study sites.** Over a 2-year period (October to December 2018, January to December 2019, and January to April 2020), small rodents were trapped for sampling in Ebonyi State, located in southeastern Nigeria. In Owo, Ondo State, located in southwestern Nigeria, small rodents were trapped from August through October 2019. The climate is tropical with two distinct seasons (rainy and dry) in both states. The average temperature ranges between 21°C and 29°C, and the humidity is relatively high in Owo, Ondo State. In Ebonyi State, the average temperature is between 25.5°C and 28.8°C. Rainfall in Owo, Ondo State, decreases in amount and distribution from the coast to the hinterland.

The samples were collected from four communities in Owo Local Government Area (LGA), Ondo State (lat 7.1989, long 5.5932), and four communities within five LGAs in Ebonyi State (Fig. 1) (lat 6.3231, long 8.1120). The communities sampled in Owo, Ondo State, were Ijebu-Owo, Isuada, Iselu, and Ehin-Ogbe. The five LGAs in Ebonyi State where traps were set are Izzi (lat 6.38577, long 8.02549), Abakaliki (lat 6.323061, long 8.112012), Ezza North (lat 6.27083, long 7.97139), and Afikpo North (lat 5.88794, long 7.95307). The communities where sampling took place in Ebonyi State were Onu-Enyim, Agbaja, Oriuzor, and Afikpo.

**Rodent trapping.** Rodents were captured using live-capture traps (H. B. Sherman Traps, Tallahassee, FL, USA) baited with a mixture of white oats, groundnut paste, and dried fish. Trapping location selection was based on communities with a history of laboratory-confirmed cases of LF. All the traps set were rebaited in the evening and checked the following morning for 4 consecutive days per week for the entire trap-setting period. The traps were set in homes, on farms, in bushes, and at dumpsites. In urban settings, traps were predominantly set in houses with a few at dumpsites. In rural areas, most traps were

set in houses, with a few on farms and in bushes. The traps in homes were set along suspected rodent paths, while the traps on farms, in bushes, and at dumpsites were set in multiple-transect lines and were also flagged for easy tracking and retrieval. Bearing in mind that some houses were large and some small, an average of three traps were set in each room, kitchen, balcony, and storage area, resulting in an average of 15 traps per household. The total number of traps set varied between 90 and 120, depending on the number of consenting houses and their sizes at each location per night of trapping. The rodent capture success was calculated according to the protocol described by Dembi et al. (13). The trap success in a location was equal to the sum of the total catch in a location divided by the number of traps set each night in that location, multiplied by the number of trap nights and multiplied by 100.

**Rodent identification, dissection, and sample collection.** Captured rodents were morphologically identified at the genus level. Biometric parameters such as weight, sex, entire length, and length of the ear, right foot, and tail were recorded. Other morphological characteristics were also noted, such as the color of the fur and its distribution/arrangement, the mammary gland arrangement and number (if a mature female rodent), the muzzle morphology, and the shape of the head. The results of species identification were confirmed through analysis of the mitochondrial genes after metagenomic sequencing of the viral RNA (5, 11) (A. N. Happi, unpublished data). The rodents were then sedated using chloroform. Blood was collected through cardiac puncture into sterile EDTA bottles. Tissue sections of approximately 0.3 to 0.5 cm$^3$ comprising brain, lung, liver, spleen, kidney, intestine, bladder, testis, embryo, and bone marrow were obtained through sterile necropsy from the euthanized rodents and preserved in TRIzol reagent (Invitrogen, CA, USA) for molecular studies.

**RNA extraction and PCR.** Total RNA was extracted from small sections of tissue and 140 $\mu$L plasma using the QIAamp viral RNA extraction kit (Qiagen, Hilden, Germany) according to the manufacturer's instructions. The samples were subjected to reverse transcriptase quantitative PCR (RT-qPCR) analysis for the detection of LASV using the Superscript III Platinum SYBR Green one-step qRT-PCR kit (Life Technologies, CA, USA) and specific primers targeting the L gene of the virus (LASV-NG forward, 5′-YAC AGG GTC YTC TGG WCG ACC-3′; LASV-NG reverse, 3′-RAT GAT GCA RCT TGA CCC AAG-5′) (14).

**Statistical analysis.** The various data were analyzed using the Student *t* test, one-way analysis of variance (ANOVA), and chi-square tests. *P* values less than 0.05 were considered statistically significant. Correlation analysis using the Pearson correlation coefficient was employed to determine the relationship between capture success and LASV prevalence.

## ACKNOWLEDGMENTS

This work was supported by BBSRC-OVEL project BB/R020116/1 and Wellcome Trust project 216619/Z/19/Z.

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
