## [Reviewer comments · Microbiology Spectrum]

Microbiology Spectrum

Increased prevalence of Lassa fever Virus positive Rodents and diversity of infected species found during human Lassa Epidemics in Nigeria

Anise Happi, Testimony Olumade, Olusola Ogunsanya, Ayotunde Sijuwola, Judih Oguzie, Chinedu Ugwu, Patricia Otuh, Seto Ogunleye, Cecilia Nwofoke, Louis Ngele, Ademola Adelabu, Oluwafemi Ojo, Samuel Okoro, Roseline Adeleye, Jonathan Heeney, Christian Happi, Nicholas OYEJIDE, and Clinton Njaka

Corresponding Author(s): Anise Happi, Redeemer's University

Review Timeline:

Submission Date:	February 1, 2022
Editorial Decision:	March 1, 2022
Revision Received:	May 18, 2022
Accepted:	June 7, 2022

Editor: Matthew Anderson

Reviewer(s): The reviewers have opted to remain anonymous.

Transaction Report:

DOI: <https://doi.org/10.1128/spectrum.00366-22>

March 1, 2022

Dr. Anise N Happi
Redeemer's University
African Centre of Excellence for Genomics of Infectious Diseases
Akoda
Ede, Osun
Nigeria

Re: Spectrum00366-22 (Increased prevalence of Lassa fever Virus positive Rodents and diversity of infected species found during human Lassa Epidemics in Nigeria)

Dear Dr. Anise N Happi:

The importance of the data in this manuscript was acknowledged by both reviewers, highlighting its potential contribution to the field. There are aspects to the data methods as noted by both reviewers that would be useful here. We also suggest reanalyzing the data in light of host sex, location, age, etc., It would also be helpful to confirm LASV by sequencing. If additional time is needed, please let me know.

Link Not Available

Sincerely,

Matthew Anderson

Journals Department
Reviewer comments:

Reviewer #1 (Comments for the Author):

The manuscript by Happi et al describes Lassa virus prevalence rates in field collected small mammals in Nigeria. Over the course of 2 years the authors collected and tested nearly 1000 small mammals from rural and urban settings in two distinct

locations of Nigeria. The findings of this study are important based on an overall lack of this type of data from Nigeria; however this reviewer recommends a more in-depth analysis and presentation of the data. Below I have attempted to point out this and other areas where I feel the manuscript could be improved.

Introduction:

1. I would add Liberia to the list of countries where LASV is endemic.
2. The sentence beginning "As of 27th August 2021, 354 confirmed cases..." lacks details. Since when (time range)? Where, across Nigeria ? in a specific region? Please modify.
3. Please be consistent with the use of LF for Lassa Fever and LASV for Lassa virus after each has been first defined in the manuscript text.

Methods:

1. What is a "Lassa season" - please define
2. Dates: IS it 2018 and 2020 or 2018 through 2020? Can the exact months of trapping be included?
3. Trapping locations: Can you provide the geographical co-ordinates in latitudes and longitudes? Overall the sites need to be better described. What is the Ecozone, climate, habitat etc. When trapping in rural settings where designated lines or schemes used or were traps just set out randomly? What was targeted in urban settings, houses? Outlying buildings, etc. Figure 1 could be expanded to include some of this information.
4. How many times per year were traps set: 4 nights every week, four nights one week a month, 4 nights every three months? What was the overall trap effort? How many traps were set at each location each night of trapping? This can then be used to calculate trap success, which should be reported if possible.
5. Rodent identification: What type of biometrics were collected from each animal? Presumably weight, sex, age (inferred by weight and/or sexual maturity)?
6. For RNA extractions, please define small sections. Is this weight (30ng?) or actual size (1cm³)?
7. Was a melt-curve analysis done as a part of the thermocycling run to differentiate positive signal from back ground noise?

Results:

1. I think this section needs to be re-evaluated. At the bare minimum it would be better to see the prevalence data presented in regards to time (year, month) of collection, LGA plus habitat (rural vs urban at a minimum) and then further analyzed by age and sex of the individual species. This would provide a far more informative summary of the data collected.
2. RT-PCR results: Please provide a more detailed summary of the results from individual tissue specimens tested. Were there different patterns of tissue positivity in *Mastomys* vs. *Crocidura* vs. *Mus* vs. *Tatera* species?
3. Sequencing confirmation should be completed and presented for at least a subset of positive rodents from each species to confirm the presence of Lassa virus. Further, sequencing data is required to definitively identify the lineage of LASV circulating in these areas. Are the viruses from the various positive species of small mammals genetically different?
4. Speciation based exclusively on physical appearance of small mammals can be error prone. The authors specify in the methods that genetic techniques for speciation were also used in the study but the results would be published elsewhere. However, given the variety of positive species identified in the current study this reviewer believes at least some genetics-based speciation is required to support their findings.
 - a. It would be helpful to know what *Mastomys* spp.
 - b. Can the authors say with certainty there were *rattus* spp. rodents that were positive based solely on appearance? Others have reported difficulty in differentiating *Rattus* and *Mastomys* based exclusively on appearance.
5. Figure 2 does not add anything to the manuscript. It should be deleted or supplemented with representative pictures of the other species captured.

Reviewer #2 (Comments for the Author):

Summary

In "Increased prevalence of Lassa fever Virus positive Rodents and diversity of infected species found during human Lassa Epidemics in Nigeria" Happi et al. describe a comprehensive survey of rodent species and their LASV positivity in two LASV-endemic regions of Nigeria. The data are comprehensive, and although the discussion is well written I feel there are some points that need to be reconsidered (over interpretation of some data regarding transmission) before publication.

Comments and Suggestions

Major:

Since a large proportion of the results describe the capture of the rodents, I feel that the way in which data points such as "capture success (%)", "prevalence (%)", and "capture frequency" are generated need to be stated clearly and within the body of the manuscript (either in the materials and methods, or in the results section). There is a reference given for capture success calculation, but I feel it would strengthen the manuscript to have all this information on hand to the reader without them having to search for further references to get the specifics. This appears to be a great data set giving valuable and interesting information, but as it is written now I am still unclear where the data for the values in Figure 1, and the data given in Results section 1a (18.6% capture success for Ondo state and 5.8% for Ebonyi State) come from for example. I also think that Figure 1 (which I

really like the style of) could be reworked to make things much clearer to the reader. If the large map of Nigeria on the left had some more detail included (some major cities, explanation of the internal boundaries marked by black lines etc) it would be helpful. Also, the switch between Ondo on the left and then Owo on the right is not immediately clear, and is not explained in the figure legend. And finally (and I might be missing this somewhere) it would be nice to have the raw numbers of rodents per capture site included in this figure as well, to go along with the capture success and prevalence? And a breakdown of types of trapping site (homes, farms, bushes, dumps) would be relevant to include in this figure as well if possible?

I also feel that the PCR positivity data needs to be expanded to offer a broader picture of the virus kinetics in these animals. Going by table 2 I think that the individual tissues were tested alone, in which case it would be incredibly useful to look more closely at what tissues types per species are infected, and whether there are any patterns between tissue tropism and species for example. This would help when topics such as transmission are described in the discussion, and give more weight to the conclusions. For example, how does virus distribution in *mastomys* spp. compare to *rattus* spp.? Are there rodent species where only a certain tissue type is routinely positive, or routinely negative compared to others?

Finally, I think some points of the discussion should be reworded to be less strong regarding conclusion on sexual transmission, vertical transmission, and interspecies transmission. Which these data certainly suggest that these route may be viable, the authors don't have the data to fully prove these aspects. A positive embryo may not be viable (what stages in pregnancy were these rodents? Is it known?). Positive testes does not correlate to virus in semen. Presence in a wide range of rodent species does not mean that the virus is spreading freely between them. I just think these caveats should be touched upon when discussion any conclusions.

Minor Issues

- Introduction: the sentence with the NCDC reference and description of "354 cases and 73 fatalities" needs a time frame (from what date until 27th August).
- Introduction: The 1st half of the second paragraph (detailing previous surveys and positivity rates) would be better presented as a table maybe?
- Figure 1 needs to be reworked for clarity (see above).
- Figure legends should be expanded to offer the reader more details. For example - what to the sub lineages refer to in figure 1? What are the units on the ruler in figure 2?

Staff Comments:

Preparing Revision Guidelines

Please return the manuscript within 60 days; if you cannot complete the modification within this time period, please contact me. If you do not wish to modify the manuscript and prefer to submit it to another journal, please notify me of your decision immediately so that the manuscript may be formally withdrawn from consideration by Microbiology Spectrum.

Response to Reviewers

We sincerely appreciate the reviewers for their constructive comments and for sharing their experience with us. It has tremendously helped improve on our manuscript.

Below are the point-by-point responses to the reviewer's comments.

Yours sincerely,

Anise Happi

Reviewer #1 (Comments for the Author):

The manuscript by Happi et al describes Lassa virus prevalence rates in field collected small mammals in Nigeria. Over the course of 2 years the authors collected and tested nearly 1000 small mammals from rural and urban settings in two distinct locations of Nigeria. The findings of this study are important based on an overall lack of this type of data from Nigeria; however this reviewer recommends a more in-depth analysis and presentation of the data. Below I have attempted to point out this and other areas where I feel the manuscript could be improved.

Introduction:

1. I would add Liberia to the list of countries where LASV is endemic.

Response: *We are thankful for this comment. We have now added Liberia with a paper reference to the list of countries where Lassa is endemic*

2. The sentence beginning "As of 27th August 2021, 354 confirmed cases..." lacks details. Since when (time range)? Where, across Nigeria? in a specific region? Please modify.

Response: *We appreciate your comment on this very important detail omitted. It has now been modified to read "The cumulative national LF situation update from January to 27th August 2021 showed 354 confirmed cases with 73 fatalities (CFR of 20.6%) reported by the Nigeria"*

3. Please be consistent with the use of LF for Lassa Fever and LASV for Lassa virus after each has been first defined in the manuscript text.

Response: *Thank you for your observation. We have now edited Lassa fever to LF consistently in manuscript.*

Methods:

1. What is a "Lassa season" - please define

Response: *Lassa season is period where there is highest number (peak period) of LF cases. It is usually between December and April each year. Although during the course of the year there are few LF cases reported as well.*

2. Dates: IS it 2018 and 2020 or 2018 through 2020? Can the exact months of trapping be included?

Response: *We appreciate this comment. We have included the exact months of trapping and it now reads “Over two year-period (October 2018 to December 2018, June 2019-December 2019), samples were collected....”*

3. Trapping locations: Can you provide the geographical co-ordinates in latitudes and longitudes? Overall the sites need to be better described. What is the Ecozone, climate, habitat etc. When trapping in rural settings where designated lines or schemes used or were traps just set out randomly? What was targeted in urban settings, houses? Outlying buildings, etc. Figure 1 could be expanded to include some of this information.

Response: *We appreciate importance of the details of the geographical coordinates and climate. We have now included them in the text. However, the ecozone and habitat is shown on the map (figure 1). Figure 1 has now been re-worked to include the boundaries, important cities, geographical coordinates, and further details on the number of rodents captured highlighted as well.*

4. How many times per year were traps set: 4 nights every week, four nights one week a month, 4 nights every three months? What was the overall trap effort? How many traps were set at each location each night of trapping? This can then be used to calculate trap success, which should be reported if possible.

Response: *Many thanks for this comment. The details on the trap nights and the total number of traps set per location per night have been included in the rodent. Yes, in calculating the trap success in a location, we sum the total of catch in a location per week divided by the number of traps set each night in that location, multiply by the number of trap night that week and multiply by 100.*

5. Rodent identification: What type of biometrics were collected from each animal? Presumably weight, sex, age (inferred by weight and/or sexual maturity)?

Response: *The biometrics collected is now added in the manuscript. But, the age of the rodent was not estimated, as it can be very subjective and was not of interest in the identification and the analysis of our results.*

6. For RNA extractions, please define small sections. Is this weight (30ng?) or actual size (1cm³)?

Response: *We sincerely appreciate this comment. During sample collection on the field, the small tissues sections were trimmed before placing them into Tryzol. However, the small tissue size was estimated to be between 0.3 to 0.5 cm³. This has now been included in the manuscript as suggested.*

7. Was a melt-curve analysis done as a part of the thermocycling run to differentiate positive signal from back ground noise?

Response: *Many thanks for this observation. A melt-curve was not done. The melt curve helped to differentiate background noise from an actual positive signal.*

Results:

1. I think this section needs to be re-evaluated. At the bare minimum it would be better to see the prevalence data presented in regards to time (year, month) of collection, LGA plus habitat (rural vs urban at a minimum) and then further analyzed by age and sex of the individual species. This would provide a far more informative summary of the data collected.

Response: *We appreciate this comment. The prevalence data presented in regards to the LGA plus habitat (rural vs urban) has already been done. Kindly see section 2c of the results. The prevalence in small rodents according to sex has now been included. It reads. “There was no significant difference in LASV positivity between male and female rodents ($P = 0.252$)”.*

For the prevalence data presented in regards to time (year) of collection has been included in the results section e.

2. RT-PCR results: Please provide a more detailed summary of the results from individual tissue specimens tested. Were there different patterns of tissue positivity in Mastomys vs. Crocidura vs. Mus vs. Tatera species.

Response: *We have now included the summary of the tissue tropism to LASV in the three most captured rodent species. See section 2g second paragraph*

3. Sequencing confirmation should be completed and presented for at least a subset of positive rodents from each species to confirm the presence of Lassa virus. Further, sequencing data is required to definitively identify the lineage of LASV circulating in these areas. Are the viruses from the various positive species of small mammals genetically different?

Response: *Many thanks for your observation and comment. Sequencing of LASV in rodent samples was not in the scope of this manuscript. This manuscript focus is to determine the prevalence of the virus among small rodents in Lassa fever hotspot of interest and determine the sample choice for detection or diagnosis using PCR.*

LASV sequencing as suggested by the reviewer is an ongoing work that might take quite some time to complete. It is also quite technically challenging to sequence and assemble complete LASV genomes for the diversity studies (vis a vis human, location, among rodent species, between organ within rodent etc...) that is being requested by the reviewer.

In our study however, we chose PCR- the gold standard for confirmation of LASV. This method is a more sensitive and specific for LASV detection, compared to sequencing. Sequencing depends significantly on the method, the platform used to generate the data, to analyze, as well as the the reference genome available.

4. Speciation based exclusively on physical appearance of small mammals can be error prone. The authors specify in the methods that genetic techniques for speciation were also used in the study but the results would be published elsewhere. However, given the variety of positive species identified in the current study this reviewer believes at least some genetics-based speciation is required to support their findings.
- a. It would be helpful to know what *Mastomys* spp.
 - b. Can the authors say with certainty there were *rattus* spp. rodents that were positive based solely on appearance? Others have reported difficulty in differentiating *Rattus* and *Mastomys* based exclusively on appearance.

Response:

- a. *From the genetics-based speciation, we found Mastomys coucha and Mastomys natalensis. We also found Mastomys mixed with other species (eg. Mastomys and Mus; M. natalensis and M. coucha etc...) from the few mastomys species sequenced. The details will be presented and discussed in the manuscript in preparation*
- b. *We confirmed the Rattus spp and Mastomys using sequencing data from mitochondrial CytC genes.*

5. Figure 2 does not add anything to the manuscript. It should be deleted or supplemented with representative pictures of the other species captured.

Response: *Many tanks for this comment. Figure 2 has now been removed as suggested by the reviewer. The representative pictures of a few of the species are just if there is need to be supplemented.*

Reviewer #2 (Comments for the Author):

Summary

In "Increased prevalence of Lassa fever Virus positive Rodents and diversity of infected species found during human Lassa Epidemics in Nigeria" Happi et al. describe a comprehensive survey of rodent species and their LASV positivity in two LASV-endemic regions of Nigeria. The data are comprehensive, and although the discussion is well written I feel there are some points that need to be reconsidered (over interpretation of some data regarding transmission) before publication.

Comments and Suggestions

Major:

Since a large proportion of the results describe the capture of the rodents, I feel that the way in which data points such as "capture success (%)", "prevalence (%)", and "capture frequency" are generated need to be stated clearly and within the body of the manuscript (either in the materials and methods, or in the results section). There is a reference given for capture success calculation, but I feel it would strengthen the manuscript to have all this information on hand to the reader without them having to search for further references to get the specifics. This appears to be a great data set giving valuable and interesting information, but as it is written now I am still unclear where the data for the values in Figure 1, and the data given in Results section 1a (18.6% capture success for Ondo state and 5.8% for Ebonyi State) come from for example. I also think that Figure 1 (which I really like the style of) could be reworked to make things much clearer to the reader. If the large map of Nigeria on the left had some more detail included (some major cities, explanation of the internal boundaries marked by black lines etc) it would be helpful. Also, the switch between Ondo on the left and then Owo on the right is not immediately clear, and is not explained in the figure legend. And finally (and I might be missing this somewhere) it would be nice to have the raw numbers of rodents per capture site included in this figure as well, to go along with the capture success and prevalence? And a breakdown of types of trapping site (homes, farms, bushes, dumps) would be relevant to include in this figure as well if possible?

Response: *We sincerely appreciate these comments:*

The way the capture success and prevalence were generated are now included in the Materials and methods and results sections of the manuscript, respectively.

The data given in the Results section 1a for the capture success in each state is calculated from the total number of traps with catch in each state for the entire trapping period divided by the total number of traps set in that state, then multiply by 100. In figure 1, the trap success results are calculated for each community separately using the same formula. In this case the total number of traps with catch in a community and the total number of traps set each community are used for calculation. Therefore adding them up cannot give the same result.

As for the map, more details have been added See new figure 1

I also feel that the PCR positivity data needs to be expanded to offer a broader picture of the virus kinetics in these animals. Going by table 2 I think that the individual tissues were tested alone, in which case it would be incredibly useful to look more closely at what tissues types per species are infected, and whether there are any patterns between tissue tropism and species for example. This would help when topics such as transmission are described in the discussion, and give more weight to the conclusions. For example, how does virus distribution in *Mastomys* spp. compare to *Rattus* spp.? Are there rodent species where only a certain tissue type is routinely positive, or routinely negative compared to others?

Response: *We are very grateful for this comment. All tissue were tested separately and each showed some degree of positivity in all rodent species. We have now included the results of the tissue type per infected species. However, it was observed that the spleen was the organ least positive for LASV in *Crociduria* species compared to *Mastomys*, and *Rattus* (Table 5) The three most common species identified.*

Finally, I think some points of the discussion should be reworded to be less strong regarding conclusion on sexual transmission, vertical transmission, and interspecies transmission. Which these data certainly suggest that these route may be viable, the authors don't have the data to fully prove these aspects. A positive embryo may not be viable (what stages in pregnancy were these rodents? Is it known?). Positive testes does not correlate to virus in semen. Presence in a wide range of rodent species does not mean that the virus is spreading freely between them. I just think these caveats should be touched upon when discussing any conclusions.

Response: *This observation is well appreciated: The language in interspecies transmission of rodents sharing the same environment has been reworded to read "All small rodent genera herewith identified showed positivity to LASV, suggesting that the virus circulates among small rodents living or sharing the same environment or ecosystem with a potential maintenance of the virus through great interspecies transmission".*

In addition, the vertical transmission has been reworded to read "The detection of LASV in embryo and foetal samples of nonspecifically different stage of gestation collected during this study is suggestive of vertical transmission..."

Furthermore, the suggested sexual transmission language has been changed and now reads "Furthermore, LASV detected in testes of male rats may suggest the presence of the virus in the semen, with the possibility of transmission by coitus.".

Minor Issues

- Introduction: the sentence with the NCDC reference and description of "354 cases and 73 fatalities" needs a time frame (from what date until 27th August).

Response: We are very thankful for this observation. This has corrected to clearly include the time frame.

- Introduction: The 1st half of the second paragraph (detailing previous surveys and positivity rates) would be better presented as a table maybe?

Response.; We appreciate your suggestion. This has been corrected as text to reduce the number of tables in the manuscript.

- Figure 1 needs to be reworked for clarity (see above).

Response: Many thanks for your observations and suggestions. Figure 1 has been reworked and included in the manuscript

- Figure legends should be expanded to offer the reader more details. For example - what to the sub lineages refer to in figure 1? What are the units on the ruler in figure 2?

Response: Many thanks for this important comment. In the Ondo State it was reported that sub-lineage IIb is predominant, while in Ebonyi it is the sub-lineage IIa that is predominant (now included in figure 1). However, we can only give history as we have not completed the sequencing the rodent LASV from our study due to some challenges. Figure 2 has been removed as it was observed that it adds no value to the manuscript.

June 7, 2022

Dr. Anise N Happi
Redeemer's University
African Centre of Excellence for Genomics of Infectious Diseases
Akoda
Ede, Osun
Nigeria

Re: Spectrum00366-22R1 (Increased prevalence of Lassa fever Virus positive Rodents and diversity of infected species found during human Lassa Epidemics in Nigeria)

Dear Dr. Anise N Happi:

Thank you for clearly addressing all reviewer comments and really strengthening the description of this study.

Your manuscript has been accepted, and I am forwarding it to the ASM Journals Department for publication. You will be notified when your proofs are ready to be viewed.

Sincerely,

Matthew Anderson
Editor, Microbiology Spectrum
